# Steroid Metabolomic Signature in Term and Preterm Infants

**DOI:** 10.3390/biom14020235

**Published:** 2024-02-17

**Authors:** Matthias Heckmann, Anna S. Runkel, Donna E. Sunny, Michaela F. Hartmann, Till Ittermann, Stefan A. Wudy

**Affiliations:** 1Department of Neonatology and Pediatric Intensive Care, University Medicine Greifswald, Sauerbruchstraße, 17475 Greifswald, Germany; anna-sophie.runkel@charite.de (A.S.R.); donnaelizabeth.sunny@med.uni-greifswald.de (D.E.S.); 2Paediatric Endocrinology & Diabetology, Laboratory for Translational Hormone Analytics, Steroid Research & Mass Spectrometry Unit, Center of Child and Adolescent Medicine, Justus Liebig University, 35392 Giessen, Germany; michaela.hartmann@paediat.med.uni-giessen.de (M.F.H.); stefan.wudy@paediat.med.uni-giessen.de (S.A.W.); 3Institute for Community Medicine, University Medicine Greifswald, 17475 Greifswald, Germany; till.ittermann@uni-greifswald.de

**Keywords:** steroids, urinary excretion rates, metabolome, preterm, term, newborn

## Abstract

Adrenal function is essential for survival and well-being of preterm babies. In addition to glucocorticoids, it has been hypothesized that C_19_-steroids (DHEA-metabolites) from the fetal zone of the adrenal gland may play a role as endogenous neuroprotective steroids. In 39 term-born (≥37 weeks gestational age), 42 preterm (30–36 weeks) and 51 early preterm (<30 weeks) infants 38 steroid metabolites were quantified by GC-MS in 24-h urinary samples. In each gestational age group, three distinctive cluster were identified by pattern analysis (*k*-means clustering). Individual steroidal fingerprints and clinical phenotype were analyzed at the 3rd day of life. Overall, the excretion rates of C_21_-steroids (glucocorticoid precursors, cortisol, and cortisone metabolites) were low (<99 μg/kg body weight/d) whereas the excretion rates of C_19_-steroids were up to 10 times higher. There was a shift to higher excretion rates of C_19_-steroids in both preterm groups compared to term infants but only minor differences in the distribution of C_21_-steroids. Comparable metabolic patterns were found between gestational age groups: Cluster 1 showed mild elevation of C_21_- and C_19_-steroids with the highest incidence of neonatal morbidities in term and severe intraventricular hemorrhage in early preterm infants. In cluster 2 lowest excretion in general was noted but no clinically unique phenotype. Cluster 3 showed highest elevation of C_21_-steroids and C_19_-steroids but no clinically unique phenotype. Significant differences in steroid metabolism between clusters are only partly reflected by gestational age and disease severity. In early preterm infants, higher excretion rates of glucocorticoids and their precursors were associated with severe cerebral hemorrhage. High excretion rates of C_19_-steroids in preterm infants may indicate a biological significance.

## 1. Introduction

Preterm infants born even at 22 weeks gestational age can survive. However, their survival rates range from 24% of infants admitted to intensive care at 22 weeks’ gestational age (GA) to 90% at 27 weeks’ GA [1,2]. These extremely preterm infants spend nearly half of their fetal life extra-uterinely, some of them starting at the end of the second trimester. The burden of critical illness is inversely proportional to gestational age. Therefore, the most immature infants suffer from the highest severity of illness. 

As a consequence, the preterm infant has to counterbalance the hormonal demands of critical illness against those of growth and organ maturation. However, preterm birth abruptly change the hormonal environment resulting in a unique hormonal milieu with respect to the function of the hypothalamic-pituitary-adrenal axis (HPAA). The human adult adrenal cortex produces mineralocorticoids, glucocorticoids (C_21_-steroids which are characterised by 2 methyl groups and a side chain with 2 carbon atoms on the sterane skeleton), and adrenal androgens (C_19_-steroids which are formed by splitting off the side chain of the C-_21_ steroids with the 2 carbon atoms). In the fetus, an additional inner zone, the fetal zone, produces extremely high amounts of C_19_-steroids, mainly dehydroepiandrosterone (DHEA) metabolites. The fetal zone involutes rapidly after birth in term infants [3]. In contrast, the fetal adrenal involution occurs not before term-equivalent age after preterm birth [4,5,6]. Particularly, in early preterm infants born with less than 30 weeks gestational age, the excretion rates of fetal zone steroids (FZS) comprise more than 90% of the total urinary steroid excretion rates [5]. Experimental data have suggested protective effects of FZSs in models of neonatal disease like cerebral ischemia, sepsis, and pulmonary hypertension [7,8,9]. In human neural stem cells derived from fetal cortex DHEA increased neurogenesis in addition to neuronal survival [10]. Our own work during the previous years has also shed light on the neuroprotective effects of selected metabolites of the fetal zone (DHEA, 16α-OH-DHEA and Adiol) in a model of hyperoxia induced cell death in immature glial cells [11]. We have shown that these steroids can distinctly influence the migration of glial progenitor cells under hyperoxic stress and that they even show sex-specific differential influence on stress response and differentiation of these cells [12]. We therefore hypothesized from the experimental data that the persistently high concentrations of FZSs might lead to a transient endogenous neuroprotection in human preterm infants [11]. 

Glucocorticoids are indispensable for cell differentiation and organ maturation in the fetus but high concentrations have deleterious effects on fetal growth and development [3]. However, cortisol release in the face of illness or stress is vital for survival postnatally. In this context, relative adrenal insufficiency is discussed as an etiology for hemodynamic instability and hypotension in the critically ill term and preterm infant [13]. However, new insights from endocrine research during critical care in adults question this approach and should be considered when analyzing the neonatal steroid metabolome. It was shown that cortisol production is only briefly increased in critical illness, followed by peripheral adaptations that maintain increased cortisol availability for target tissues without continued need for increased ACTH-driven cortisol production and secretion [14]. These adaptations comprises suppressed cortisol breakdown in liver and kidney [15]. Furthermore, plasma concentrations and binding-affinity of cortisol carrier proteins are reduced, resulting in higher free cortisol concentrations [16,17]. In prolonged critical illness a decrease of plasma total and free cortisol concentrations and delayed suppressed ACTH response to CRH test are indicative of central adrenal insufficiency that may develop over weeks [18]. Therefore, cortisol inactivation and the time course of illness should be considered when analyzing the steroid metabolome in preterm infants.

In the setting of the extra-uterinely maturing preterm infant during critical illness, non-invasive methods are required. Gas chromatography-mass spectrometry (GC-MS) allows for the most comprehensive approach characterizing the complex urinary steroid metabolome [19]. Furthermore, a cohort of neonates covering the whole range of gestational age and comprising detailed characterization of the clinical phenotype to consider illness severity is required.

Our aim was to characterize changes in the pattern of the steroid metabolome of preterm and term infants to analyse the association of illness severity, gestational age and time course with the steroid metabolomic signature. A particular focus is on fetal zone steroid and glucocorticoid metabolites.

## 2. Patients and Methods

### 2.1. Patients

The study population consisted of 2 cohorts of preterm infants (gestational age < 30 and 30–36 weeks) enrolled originally in a prospective longitudinal study to investigate adrenal steroid production by gas chromatography mass spectrometry (GC–MS) in relation to illness and postnatal growth [19,20]. Term infants with a gestational age > 37 weeks were enrolled separately from March 2009 until July 2010 [21]. The database of these three cohorts was used to re-analyse the steroid metabolome with respect to single steroid metabolites.

The study was approved by the ethics committee of Justus Liebig University of Giessen (reference number 04/02). The attending physician informed the parents about the study to obtain written parental consent. We did not expect a major selection bias because the overall acceptance of the study was good because there was no intervention or invasive procedure. Only the diapers were collected. Therefore, the total number of patients screened and parental refusals were not recorded with the exception of the early preterm group because this gestational age group represented only 10 percent of neonatal patients treated. The 67 preterm infants with a gestational age of less than 30 weeks were admitted to our neonatal unit between July 2001 and September 2002. Sixty-one of them matched the study criteria. Reasons for exclusion were postnatal steroid therapy (*n* = 5) and chromosomal aberration (trisomy 21, *n* = 1).

Exclusion criteria included a family history of adrenal illnesses, chromosomal disorders and postnatal steroid therapy. Gestational age was determined using the expanded Ballard score and/or obstetrical dating.

Term infants were classified as being well when they received only routine care after delivery in the well-baby nursery of the department of obstetrics and were not admitted to the department of neonatology in the children’s hospital.

Term infants were classified as being ill when one or more of the following diseases were diagnosed: respiratory illness (transient tachypnea of the newborn, respiratory distress syndrome, and congenital pneumonia), early-onset infection (C-reactive protein > 10 mg/L and symptoms of an infection during the first 72 h of life) or hypoglycemia (blood glucose level < 2.5 mmol/L (45 mg/dL).

Preterm infants were classified as being well when they had no signs of infection, did not receive treatment with surfactant or inotropes, and had serum glucose concentrations > 2.2 mmol/L (40 mg/dL) [19]. The management of preterm infants and ill term infants had to avoid hypoxemia and hyperoxemia. Therefore, oxygen saturation as measured by pulse oximetry was kept in the target range of 88% to 95% in both, preterm and term neonates who were treated with supplemental oxygen. Mechanical ventilation due to apnea was not considered an exclusion criterion for being well in order to include even extremely immature preterm infants. 

Ill preterm infants suffered from one or more of the following diseases [19]: respiratory distress syndrome treated with surfactant, infection at birth (C-reactive protein > 10 mg/L and symptoms of an infection during the first 72 h of life), hospital infection (C-reactive protein > 10 mg/L, and symptoms of an infection after the first 72 h of life), ventricular hemorrhage more than II° (intraventricular hemorrhages are categorized by severity from I–IV; II refers to a bleeding which is found in the ventricles with no change in ventricle size; [22]).

To assess the severity of illness, all infants were scored using the Score for Neonatal Acute Physiology (SNAP) on each day when CPRs were determined [23].

A prenatal betamethasone therapy was recorded as being complete if 2 doses of 12 mg of betamethasone were given to the mother more than 24 h ante partum. 

### 2.2. Urine Collection Procedure

Twenty-four–hour urine samples were collected frequently in the first week of life, namely, at the first, second, third, and fifth day of life, because severity of illness was expected to be the highest in this period. Thereafter, 24-h urine collections were made at weekly intervals during the first month of life and then monthly up to discharge. The urine collection procedure has been described in detail previously [19]. In brief, urine was collected in special cellulose-only diapers in 2 sizes (weight 16 g in preterm infants below 2300 g, 22 g above) (Pampers^®^, Procter & Gamble, Schwalbach, Germany). Weighing the diaper before and after urine collections allowed exact calculation of 24-h urine output. Diapers were changed as necessary or at 4 to 8 h at the latest. To reduce contamination with meconium or stool to a minimum, thin gauze was placed between the baby’s skin and the surface of the inner side of the diaper, thus allowing urine to pass through the gauze and, at the same time, withholding meconium or stool, which then could easily be separated from the diaper. Samples from infants with diarrhea were omitted. Urine was extracted by compressing the diaper using a hydraulic press applying a maximum of 120 kilopascal (kPa)/cm^2^. After centrifugation, the collected urinary specimens were stored at −80 °C until analysis by GC–MS.

### 2.3. Laboratory Analyses

Urinary steroid profiles were determined by GC–MS analysis according to our procedure described recently [19]. We measured 38 metabolites by selected ion monitoring in term and preterm infants at the 3rd day and 2nd week of life (Appendix A). 

Daily urinary steroid excretion rates were corrected for body weight. The ratio of 11-OH/11-OXO metabolites, as an estimate of 11ß-hydroxysteroid dehydrogenase (11ß-HSD) activity (THF + F+ 6ß-OH-F + 20α-DHF)/(THE + α-Cl + 6α-OH-THE + 1ß-OH-THE + ß-Cl + 6α-OH-α-Cl +6α-OH-ß-Cl + 1ß-OH-ß-Cl) was calculated. Creatinine (mg/day) was determined to verify completeness of 24-h collection.

### 2.4. Statistical Analysis of Metabolomic Data

The concept of individual steroid fingerprinting and “steroid metabolic disease signature” was applied based on quantitative GC-MS data of urinary steroids [20,21]. Steroid metabolites quantities were *z*-transformed. STATA version 17.0 (StataCorp. LLC, USA) was used to generate the clusters of patients and subsequent statistical analysis. STATA generated three unique clusters by taking as an input steroid metabolites excretion profiles as an input and by invoking the k-means clustering algorithm, where 3 cluster was chosen by visual inspection of the heatmap at day three (Figure 1a). A subsequent principal component analysis (PCA) has been performed (Figure 1b). Z-transformed quantities of 38 steroid metabolites were used as features for this analysis. Steroidal signatures and clinical data of patients in each cluster were analyzed, and ANOVA was utilized to assess the clinical differences between clusters. Timing was chosen to maximize the number of urine samples available during the first week of life when disease severity was high and a later time point after recovery from acute illness [19,20]. Hence, the time points 3rd day of life and 2nd week of life were selected. However, this analysis is limited to preterm infants, as the term infants were predominantly discharged before the 2nd week of life. *p*-value < 0.05 was considered statistically significant. When comparing daily excretion rates of single steroid metabolites between clusters, multiple testing was corrected by the Bonferroni adjustment. Biomarker analysis was performed by classic receiver operating characteristic (ROC) curve analysis.

## 3. Results

Urinary steroid metabolome data sets were available in 39 term infants at day 3 and in 42 preterm as well as 51 early preterm infants at day 3 and in the 2nd week of life. Illness severity differed impressively between these three gestational age groups (Table 1, Table 2, Table 3, Table 4, Table 5, Table 6, Table 7, Table 8 and Table 9). Supplemental Appendix A describes the daily urinary steroid metabolome with their relative contribution to the total circulating steroid pool depending on gestational age. 

### 3.1. Term Infants

The ‘steroid metabolomic signatures’ of the 3 clusters generated in term infants are shown in Figure 2 and their descriptive data are presented in Table 1, Table 2 and Table 3.

Gestational age and birthweight were not significantly different between the clusters. Cluster 1 patients had mild elevation of C_21_-steroids (glucocorticoid precursors, cortisol, and less cortisone metabolites). At the clinical level, they all were classified as being ill, suffering mainly from infection and respiratory distress syndrome. In contrast, lowest excretion of steroids without a specific pattern was observed in cluster 2, where all newborns were classified as well. Therefore showing significant differences when compared to cluster 1 patients with respect to SNAP score, rate of infection and respiratory distress. In cluster 3, 75% of patients were classified as well and no differences were found when compared to the other clusters at the clinical level. At the same time, highest elevation of C_21_-steroids (glucocorticoids: both cortisol and cortisone metabolites), 17-OH-pregnenolone metabolites and particularly C_19_-steroids (DHEA-metabolites) with the highest excretion rates were observed (Table 1, Table 2, Table 3 and Appendix A) in this cluster. 

Additionally, no differences were found between clusters in the ratio of 11-OH/11-OXO metabolites, as an estimate of 11ß-hydroxysteroid dehydrogenase (11ß-HSD) activity (Table 3).

Appendix A gives a graphical representation of the distribution of daily urinary steroid metabolome in term infants. The urinary excretion rates of the glucocorticoid precursors, progesterone, 17-hydroxyprogesterone and 11-deoxycortisol and cortisol as well as cortisone metabolites (except 6α-OH-ß-Cl) were generally low (<99 μg/kg body weight/d) whereas the FZS represented the majority of urinary steroid excretion.

### 3.2. Preterm and Early Preterm Infants 

The metabolomic signature generated by clustering the steroid metabolome as a function of gestational age (preterm infants 30–36 weeks gestational age vs. early preterm infants < 30 weeks) and postnatal age (3rd day of life vs. 2nd week of life) in the preterm group is shown in Figure 3. Their descriptive data are presented in Table 4, Table 5 and Table 6 (preterm) and Table 7, Table 8 and Table 9 (early preterm).

In preterm infants (30–36 weeks), gestational age and birthweight were not significantly different between the clusters. At day 3, a general mild elevation of C_21_-and C_19_-steroids were found in cluster 1. Cluster 2 patients had the lowest excretion of steroids without any specific pattern. The highest excretion rates of steroid metabolites were found in cluster 3, i.e., highest elevation of C_21_-steroids (17-hydroxyprogesterone, and cortisone metabolites), 17-hydroxypregnenolone metabolites and C_19_-steroids (DHEA-metabolites) (Table 6). At the clinical level, all clusters comprise well and ill preterm infants without any significant differences in severity or type of disease. In the second week, considerably overlap in the metabolic clusters was observed while the median (min-max) SNAP score decreased from 5 (0–15) at the 3rd day to 2 (0–7) in the 2nd week. Differences were seen mainly in C_19_-steroids with higher excretion rates in cluster 1 and 3 compared to cluster 2.

In early preterm infants (<30 weeks), gestational age and birthweight were not significantly different between the clusters. At day 3, patients of the largest cluster 1 showed mild elevation of C_21_-steroids, and C_19_-steroids. At the clinical level, cluster 1 patients had the highest incidence of severe IVH compared to cluster 2 and 3 (*p* = 0.08). The odds ratio for severe IVH was 4.33 (CI 0.46–40.5; *p* = 0.198) for cluster 1 versus cluster 2. Cluster 2 patients had the lowest excretion of steroids without specific pattern. The highest elevation of C21-steroids (17-hydroxypregnenolone and 17-hydroxyprogesterone metabolites, and cortisone metabolites) and C19-steroids were found in cluster 3. Except for a trend to a higher incidence of severe IVH in cluster 1, no significant differences in severity or type of disease were found between the clusters. In the second week, the metabolic clusters showed overlap but to a lesser extent compared to the more mature preterm infants as the median (min-max) SNAP score decreased from 15 (2–33) at the 3rd day to 5 (2–28) in the 2nd week.

In preterm and early preterm infants, there was a shift to higher excretion rates of FZS compared to term infants but only minor differences in the distribution of C21-steroids between the gestational age groups (Appendix A).

### 3.3. Biomarker Selection

Highest severity of illness was found in early preterm infants. Additionally, an accumulation of severe IVH was found in cluster 1. Therefore, selection of biomarker was performed in the early preterm group with respect to differentiation in the categories ill versus well infants and severe cerebral bleeding at day 3. With respect to differences seen in Table 9 and to the level of excretion rates (Appendix A) we selected 3 glucocorticoid-metabolites and 2 DHEA-metabolites for analysis. However, ROC analysis did not identified any of the selected metabolites to discriminate between ill and well early preterm infants(Number of observations = 51): ß-Cl (area under ROC curve = 0.4704), 6α-OH-α-Cl (0.5370), 6α-OH-ß-Cl (0.6093), 16α-OH-DHEA (0.5167), and A5-3ß16α17ß18tetrol (0.5500). Furthermore, ROC analysis did not identified any of the selected metabolites to discriminate between severe IVH (>2) and less severe or no IVH in early preterm infants (Number of observations = 51): ß-Cl (area under ROC curve = 0.6104), 6α-OH-α-Cl (0.5357), 6α-OH-ß-Cl (0.5065), 16α-OH-DHEA (0.5812), and A5-3ß16α17ß18-tetrol (0.4675).

## 4. Discussion

Here, the concept of individual steroid fingerprinting and “steroid metabolic signature” was applied based on quantitative GC-MS data of urinary steroids in neonates early in life during acute illness [24]. Since gestational age is the most important factor affecting severity of disease and organ maturity three gestational age groups were analyzed separately. In each gestational age group, metabolic data analysis identified three distinctive clusters with characteristic ‘steroid metabolomic signatures’. Despite the differences in gestational age and associated differences in disease severity comparable metabolic pattern were found to be comparable after *z*-transformation between gestational age groups. 

Cluster 1 that comprises 41% of all term, preterm and early preterm infants, showed mild elevation of C21- and C19-steroids. In term infants the highest incidence of neonatal morbidities was found in this cluster with 100% of infants classified as ill in contrast to the other term clusters that mainly included well infants (79%). Within the early preterm group, cluster 1 exhibits the highest occurrence of severe cerebral bleeding, alongside heightened levels of cortisol precursors and cortisone metabolites. Studies have previously reported an association between cortisol serum concentrations, cortisol production rates and the occurrence of high-grade intraventricular hemorrhage in preterm infants [19,25,26,27]. The elevated cortisol concentrations and glucocorticoid metabolites levels experienced by extremely premature infants, and at the same time signifies the functioning of the adrenal stress response even in early preterm infants. Here, the metabolome analysis provides a more comprehensive and detailed perspective on these findings. 

The level of adrenal stress response in the early preterm group (3rd day) appears comparable to the term group in cluster 1, despite the considerably higher severity of illness in early preterm infants. Notably, cortisone metabolites are elevated, suggesting a potential regulation of the adrenal stress response through cortisol inactivation. This could serve to protect against excessively high cortisol concentrations which may occur in case of high illness severity in early preterm infants. Conversely, in the preterm group (30–36 weeks gestational age), there was no distinct clinical phenotype corresponding to the observed metabolic changes in cluster 1, as seen in term and early preterm infants. 

C19-steroids were mildly elevated in cluster 1. The adrenal fetal zone is responsible for producing the adrenal C19-steroid dehydroepiandrosterone (DHEA) in large amounts, which is converted to estriol by the fetal-placental unit [28,29]. Beside their role as precursor for the placental estrogen synthesis during fetal life, DHEA and other androgens also play a role in critical illness. In adults, dehydroepiandrosterone sulphate (DHEAS) concentrations were found to be low, along with high cortisol concentrations, particularly in adult non-survivors of septic shock associated with trauma [30,31].In contrast, a dissociation in excretion rates of glucocorticoids and C-19 steroids was not observed in any of the clusters in our study. Furthermore, the catabolic response to severe injury was accompanied by acute and sustained androgen suppression over weeks [32,33]. DHEA has immune stimulatory and anti-glucocorticoid effects, and human supplementation studies in post-menopausal females, older adults, or individuals with adrenal insufficiency have shown that restoring the cortisol: DHEAS ratio improves wound healing, mood, bone remodeling and psychological well-being [34,35]. In view of these observations, recently, a randomized study investigating DHEA supplementation in adult patients with acute trauma was initiated [36]. Furthermore, oxandrolone, a synthetic analog, 17-alpha-methyl derivative of testosterone, is being considered as a therapeutic agent in adults with severe burns [37]. Unlike FZS, that comprises 90% of the adrenal steroid synthesis until the 5th month of life in preterm infants; the serum concentrations of DHEA/DHEAS undergo a steady age-related decline after the third decade of life, such that by 70 years of age, levels are 20–30% of those achieved in early adulthood [5,35]. We therefore speculated that the persistently high concentrations of FZSs might lead to a transient endogenous neuroprotection in preterm infants [11]. 

Cluster 2 shows lowest excretion rate of C21- and C19-steroids in general and comprises 35% of all infants from term to early preterm. While the corresponding phenotype in this cluster primarily includes well or moderately sick infants of the term and preterm group, 78% of the early preterm infants with the metabolic pattern of cluster 2 were severely ill. Here one would have expected an adrenal stress response comparable to cluster 1 or 3. However, even the excretion rates of C19-steroids were also low in this cluster. The development and function of the fetal adrenals is tightly regulated, principally by adrenocorticotropic hormone (ACTH) [38]. However, ACTH and concentrations of FZS are inversely proportional to each other in early preterm infants indicating that ACTH might not be the sole regulator of the fetal zone [6]. It is unclear why early preterm infants of cluster 2 showed general low steroid excretion rates in face of severe illness. Perhaps these infants may benefit from precision medicine in the form of targeted steroid supplementation based on steroid metabolome analysis. Steroid metabolome analysis may be a new approach to guide therapy in early preterm infants since random cortisol concentrations do not correlate with hydrocortisone therapy [27,39]. However, low cortisol concentrations alone did not identify the infants at highest risk for adverse outcomes. In contrast, high cortisol values were associated with increased morbidity and mortality rates [39]. Particularly the administration of hydrocortisone to prevent bronchopulmonary dysplasia (BPD) is meanwhile investigated in a very large number of preterm infants. Two multicenter randomized controlled trials starting hydrocortisone in the second week of life resulted in no beneficial effects but starting earlier resulted in reduction of BPD in a third trial [40,41,42]. Timing, as well as dosing of hydrocortisone differed between the trials and varied between low dose (supplementation of cortisol) and anti-inflammatory, pharmacological higher doses [40,41,42]. In our group of early preterm infants, none of them had moderate or severe BPD or received postnatal therapy with glucocorticoids. Finally, one can speculate that these even the low excretion rates in cluster 2 are adaptive and that treatment with stress-doses of hydrocortisone may negatively interfere with these adaptive changes.

Cluster 3 shows highest excretion rate of C21- and C19-steroids in general and comprises 24% of the neonates investigated in this study. The corresponding phenotype in this cluster primarily includes mainly well infants of the term and preterm group. When comparing the clinical phenotype of term and preterm infants with respect to cluster 2 with the lowest excretion rate and cluster 3, no significant differences were noted. In early preterm infants, the highest proportion of well infants (46%) was found in cluster 3. In contrast to cluster 1 with elevated cortisol metabolites, cortisone metabolites and FZS were highest in cluster 3 of the early preterm group. This could not be explained by a temporal shift of disease severity with more earlier maximum of disease severity in cluster 1 because SNAP at day 3 was lower in early preterm infants than at day 2, because there were no differences in temporal distribution of maxima off illness severity measured by SNAP between the clusters in the early preterm group. 

In the second week of life, severity of illness measured by SNAP decreased in the majority of preterm and early preterm infants. At the same time, the metabolic clusters converge. The substantial overlap in metabolic clusters in the second week suggests a lack of clear differentiation in the steroid metabolomic profiles among the different groups of infants. Although there were significant differences in steroid metabolism between clusters in the immediate postnatal phase at day 3, these are only partly associated with the clinical phenotype, i.e., disease severity in term infants and incidence of severe cerebral hemorrhage in the early preterm group. Infants at 7–8 months of age show an significant several fold increase in urinary cortisol metabolites as measured by GC-MS after surgery and similar increase has also been observed after trauma in adults [14,43]. However, only a few studies are available where absolute measurements have been performed by using GC-MS. With respect to different stressors, cortisol concentrations were determined in children and adolescents showing increased cortisol values which varied across a range of different acute illnesses [44]. This might be the case in preterm infants also but could not be detected in or study except for severe cerebral hemorrhage. 

The strengths of our study are that our cohort covers the whole range of gestational age from the boarder of viability until term with detailed description of the clinical phenotype. Furthermore, we were able to provide steroid determination in 24-h urinary specimens as a noninvasive assessment of steroid metabolome. Only the highly specific and nonselective analytical technique of GC–MS allows profiling of urinary FZSs in premature infants since there is no commercially available immunoassay for measuring these typical neonatal steroids [5,19,45]. However, the complexity of data interpretation of the steroid metabolome is exacerbated by the vulnerability of the preterm infant and the numerous exposures during early life. Prematurity is by definition an abnormal condition which is the result of various prenatal complications and subsequent higher incidence of one or more severe neonatal diseases. Therefore, a study group of particularly early preterm infants will always be much more heterogeneous than a group of term infants. Although, a quite large cohort of 132 newborns was analyzed here, of whom 51 were born as early preterm infants of less than 30 weeks gestational age, the statistical analysis is additionally limited due to the wide distribution in concentrations of the steroid metabolites. Biomarker analysis failed to identify a metabolite which discriminate between well and ill early preterm infants and those with and without severe intracranial hemorrhage. Therefore, recognizing the sample size limitation, we do not claim this model to have a predictive value but at the same time, hope that we and others would be able to expand the data to allow it in the future. The study does not provide conclusive evidence of the practical utility of this approach in a clinical setting. Furthermore, analysing steroid metabolites exclusively on the 3rd day and 2nd week of life might overlook critical variations at other times.

## 5. Conclusions

In conclusion, while the study explores the relationship between steroid metabolites and disease severity, it acknowledges the complexity of the interplay between various factors. It recognizes that gestational age and disease severity alone do not account for all the variations observed in the metabolome clusters. However, in early preterm infants with the highest illness severity, higher excretion rates of glucocorticoids and their precursors were associated with an accumulation of severe cerebral hemorrhage. Neither the highest excretion rates nor general low excretion were associated with a special clinical phenotype. Furthermore, analysis of selected steroid metabolites failed to identify reliable biomarkers for distinguishing between ill and well early preterm infants. Particularly in the early preterm infants with high illness severity and burden of long-term effects, clustering the neonatal steroid metabolome needs further evaluation and may serve as a basis to develop cluster-specific therapeutic strategies.

## Figures and Tables

**Figure 1 biomolecules-14-00235-f001:**
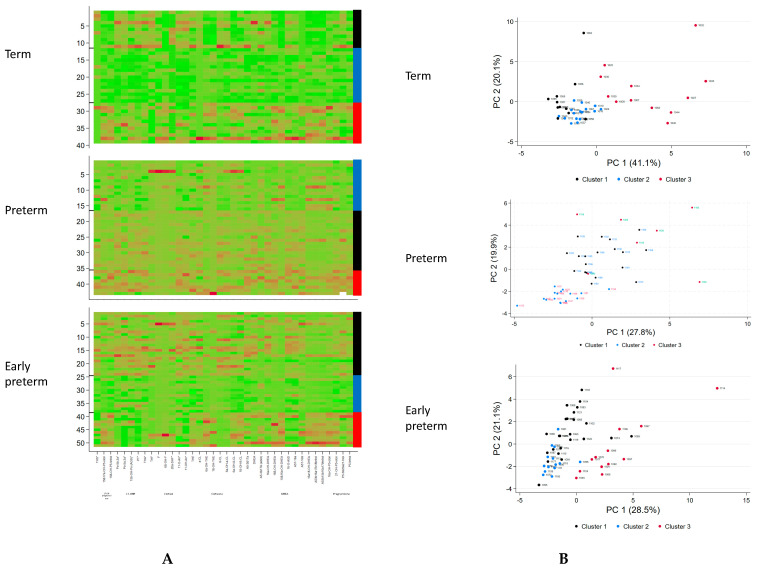
Urine steroid metabolites excretion analysis in term, preterm and early preterm infants. (**A**) Heatmap and metabolite clusters: the rows represent the patients and the columns the *z*-transformed 24-h excretion rates of the 38 steroid metabolites; the colors in the heatmap are red to green indicating high to low excretion rates of the metabolites. Columns at the margin indicate the clusters assigned (black cluster 1, blue cluster 2, red cluster 3). (**B**) Principal component (PC) analysis. Each dot represents one of the 39 (term)/42 (preterm)/51 (early preterm) samples projected on the principal plane formed by the first and second principal axes. The dots are colored according to the subject’s classification group.

**Figure 2 biomolecules-14-00235-f002:**
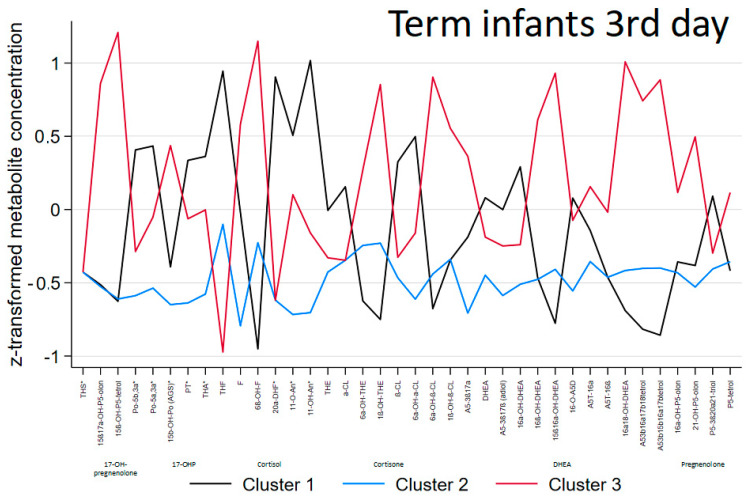
Metabolomic signatures of clustered steroid metabolome in term infants at the 3rd day of life. The *x*-axis includes 38 steroid metabolites. The *y*-axis indicates the *z*-transformed 24-h urinary excretion rates of steroid metabolites. Median values of subjects in each cluster are connected by colored lines according to cluster allocation.

**Figure 3 biomolecules-14-00235-f003:**
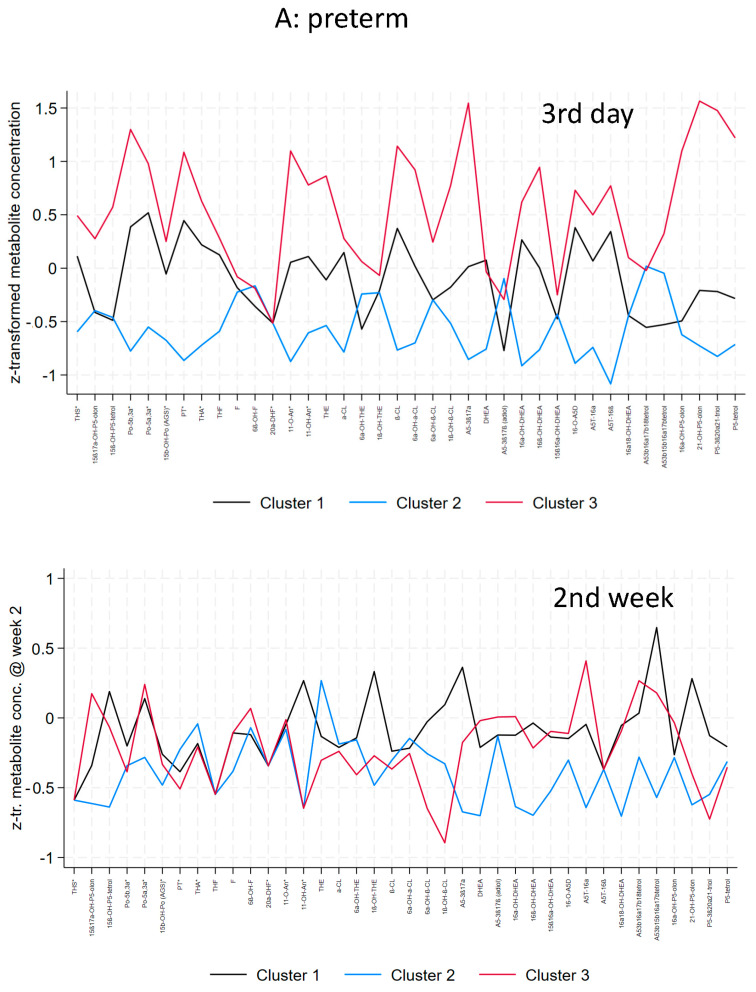
Metabolomic signatures of clustered steroid metabolome as a function of gestational age (preterm infants 30–36 weeks gestational age (**A**) vs. early preterm infants < 30 weeks (**B**) and postnatal age (3rd day of life vs. 2nd week of life). The x-axis includes 38 steroid metabolites. The y-axis indicates the *z*-transformed 24-h urinary excretion rates of steroid metabolites. Median values of subjects in each cluster are connected by colored lines according to cluster allocation.

**Table 1 biomolecules-14-00235-t001:** Distinctive metabolic characteristics of the generated 3 clusters in term infants at the 3rd day of life.

	Cluster 1	Cluster 2	Cluster 3
*n*			
Distinctive metabolic characteristics	Mild elevation of C21-steroids (glucocorticoid precursors, cortisol metabolites and less cortisone metabolites) and C_19_-steroids (DHEA-metabolites)	No specific abnormality, lowest excretion in general	Highest elevation of C21-steroids (glucocorticoids: both cortisol and cortisone metabolites), 17-OH-pregnenolone metabolites and C_19_-steroids (DHEA-metabolites)

**Table 2 biomolecules-14-00235-t002:** Clinical phenotype of well and ill term infants according to cluster division (median (min-max)) at the 3rd day of life.

	Cluster 1		Cluster 2		Cluster 3		*p*-Value
Group (*n*)	Well (0)	Ill (11)	Well (13)	Ill (3)	Well (9)	Ill (3)	
GA		39 (38–41)	40 (37–42)	41 (40–41)	39 (37–41)	39 (38–39)	0.08
BW		3050 (1860–3750)	3260 (2710–4280)	3620 (3120–3700)	3410 (2910–3930)	3310 (2710–3650)	0.25
Sex (female)		4 (11)	8/13	2/3	6/9	1/3	
SGA		5/11	2/13	0	0	1/2	0.05
Apgar 5 min		10 (8–10)	10 (9–10)	10 (8–10)	10 (7–10)	8 (8–10)	0.76
C-section		2/11	7/13	0	6/9	1/3	0.14
SNAP		2 (0–19)	0	2 (0–7)	0	5 (1–6)	0.002^C1>2^
Infection		8/11	0	1/3	0	2/3	0.0005^C1>2^
RDS		7/11	0	0	0	2/3	0.0005^C1>2^
CHD		1/11	0	1/3	0	1/3	0.96
Hypo-glycemia		1/11	0	1/3	0	1/3	0.96

GA gestational age; BW birth weight; SGA small for gestational age (BW < 10th percentile); SNAP score for acute neonatal physiology; RDS respiratory distress syndrome; CHD congenital heart defect.

**Table 3 biomolecules-14-00235-t003:** Z-transformed daily excretion rates of steroid metabolites of well and ill term infants according to cluster division (Mean (and SD)) and the 3rd day of life.

	Cluster 1	Cluster 2	Cluster 3	*p*
*n*	11 (28.2%)	16 (41.0%)	12 (30.8%)	
THS	0.397 (1.213)	−0.206 (0.612)	−0.088 (1.174)	0.293
15ß,17OH-P5o	−0.440 (0.362)	−0.502 (0.284)	1.073 (1.186)	<0.001
P5-tetrol-15ß	−0.414 (0.655)	−0.557 (0.338)	1.122 (0.955)	<0.001
17OHPo	0.889 (1.356)	−0.561 (0.191)	−0.067 (0.693)	<0.001
17OHPo-5α	0.538 (0.580)	−0.565 (0.432)	0.259 (1.438)	0.007
15ß,17OHPo	0.220 (1.370)	−0.604 (0.331)	0.603 (0.790)	0.003
PT	0.866 (1.403)	−0.630 (0.155)	0.046 (0.551)	<0.001
THA	0.800 (1.335)	−0.634 (0.285)	0.112 (0.697)	<0.001
THF	1.100 (0.835)	−0.276 (0.535)	−0.641 (0.814)	<0.001
F	0.153 (0.946)	−0.588 (0.520)	0.644 (1.136)	0.002
6ß-OH-F	−0.847 (0.246)	−0.191 (0.481)	1.031 (1.074)	<0.001
20α-DHF	0.624 (1.257)	−0.442 (0.486)	0.018 (1.011)	0.020
11-O-An	0.658 (0.873)	−0.674 (0.394)	0.295 (1.167)	<0.001
11-OH-An	0.938 (1.003)	−0.677 (0.267)	0.043 (0.946)	<0.001
THE	0.508 (1.352)	−0.472 (0.187)	0.164 (1.065)	0.030
α-Cl	0.315 (0.431)	−0.332 (0.039)	0.154 (1.732)	0.212
6α-OH-THE	−0.495 (0.470)	−0.266 (0.281)	0.809 (1.443)	0.001
1ß-OH-THE	−0.740 (0.504)	−0.190 (0.442)	0.932 (1.187)	<0.001
ß-Cl	0.828 (1.546)	−0.458 (0.130)	−0.149 (0.495)	0.002
6α-OH-α-Cl	0.600 (1.130)	−0.547 (0.502)	0.179 (1.058)	0.007
6α-OH-ß-Cl	−0.551 (0.579)	−0.368 (0.557)	0.996 (1.069)	<0.001
1ß-OH-ß-Cl	−0.140 (0.812)	−0.383 (0.525)	0.639 (1.342)	0.020
A5-3ß,17α	0.336 (1.349)	−0.575 (0.345)	0.459 (0.902)	0.007
DHEA	0.534 (1.597)	−0.390 (0.273)	0.030 (0.725)	0.057
Adiol	0.415 (1.217)	−0.266 (0.836)	−0.026 (0.938)	0.223
16α-OH-DHEA	0.431 (1.173)	−0.429 (0.264)	0.176 (1.273)	0.065
16ß-OH-DHEA	−0.377 (0.303)	−0.453 (0.463)	0.950 (1.286)	<0.001
15ß,16α -OH-DHEA	−0.781 (0.118)	−0.322 (0.209)	1.146 (1.090)	<0.001
16-O-A5D	0.314 (0.857)	−0.502 (0.373)	0.382 (1.404)	0.028
A5T-16α	−0.012 (0.510)	−0.359 (0.415)	0.490 (1.591)	0.081
A5T-16ß	−0.462 (0.000)	−0.189 (0.637)	0.676 (1.462)	0.011
16α,18-OH-DHEA	−0.639 (0.205)	−0.397 (0.208)	1.114 (1.168)	<0.001
A5-3ß16α17ß18-tetrol	−0.672 (0.345)	−0.274 (0.399)	0.981 (1.232)	<0.001
A5-3ß15ß16α17ß-tetrol	−0.729 (0.368)	−0.269 (0.464)	1.026 (1.119)	<0.001
16OHP5o	−0.336 (0.262)	−0.234 (0.439)	0.620 (1.587)	0.030
21OHP5o	0.001 (0.874)	−0.521 (0.270)	0.694 (1.320)	0.004
P5-3ß20α21-triol	0.506 (1.282)	−0.438 (0.287)	0.121 (1.138)	0.044
P5-tetrol	−0.057 (1.025)	−0.375 (0.474)	0.551 (1.286)	0.047
11ß-HSD *	−0.136 (0.752)	−0.017 (1.232)	0.147 (0.910)	0.800

* ratio of 11-OH/11-OXO metabolites.

**Table 4 biomolecules-14-00235-t004:** Distinctive metabolic characteristics of the generated 3 clusters in preterm infants at the 3rd day and 2nd week of life.

	Cluster 1	Cluster 2	Cluster 3
*n*			
Distinctive metabolic characteristics (day 3)	General mild elevation of C_21_-and C_19_-steroids (no specific pattern)	No specific abnormality, lowest excretion in general	Highest elevation of C21-steroids (17-OH-progesterone, cortisone metabolites), 17-OH-pregnenolone metabolites and C_19_-steroids (DHEA-metabolites)
Distinctive metabolic characteristics (week 2)	General mild elevation of C_21_-and C_19_-steroids (no specific pattern)	No specific abnormality, lowest excretion in general	Mild elevation of 17-OH-progesterone metabolites, low cortisone metabolites, and mild elevation of C_19_-steroids but considerably lower than cluster 3 at day 3

**Table 5 biomolecules-14-00235-t005:** Clinical phenotype of well and ill preterm infants according to cluster division (median (min-max)).

	Cluster 1		Cluster 2		Cluster 3		*p*-Value
Preterm	Well (*n* = 14)	Ill (*n* = 5)	Well (*n* = 10)	Ill (*n* = 6)	Well (*n* = 5)	Ill (*n* = 2)	
GA	32 (30–35)	32 (31–33)	33 (32–36)	34 (31–35)	34 (32–35)	32.4, 33.3	0.66
BW	1790 (1120–2600)	1730 (1150–2250)	2130 (1700–2690)	1630 (1470–2630)	1800 (1220–2455)	2160, 2320	0.52
Sex (female)	3/14	2/5	3/10	1/6	3/5	1/2	
SGA	0/14	1/5	2/10	2/6	0/5	0/2	0.11
Apgar 5 min	8 (7–10)	9 (2–10)	9 (7–10)	8 (4–9)	9 (9–10)	9,10	0.05
C-section	13/14	4/5	9/10	5/6	0/5	1/2	1.00
pH	7.35 (7.21–7.39)	7.3 (7.27–7.37)	7.3 (7.23–7.41)	7.32 (7.23–7.62)	7.3 (7.3–7.36)	7.33, 7.36	0.88
Prenatal Steroids	5/14	2/5	6/10	1/6	1/5		0.94
SNAP D3	2 (0–7)	9 (7–12)	5 (2–7)	6 (3–15)	2 (0–5)	6	0.92
SNAP W2	2 (0–6)	4 (1–6)	1 (0–6)	2 (1–7)	1 (0–2)	1,2	0.10
RDS		1/5	4/10	1/6	2/5		0.63
Infection	0/14	2/5	1/10	3/6	0/5		0.24

**Table 6 biomolecules-14-00235-t006:** Z-transformed daily excretion rates of steroid metabolites of well and ill preterm infants according to cluster division (Mean (and SD)) and the 3rd day of life.

	Cluster 1	Cluster 2	Cluster 3	*p*
*n*	19 (45.2%)	16 (38.1%)	7 (16.7%)	
THS	0.184 (0.710)	−0.576 (0.916)	0.760 (1.302)	0.005
15ß,17OH-P5o	−0.308 (0.397)	−0.030 (1.259)	0.809 (1.192)	0.038
P5-tetrol-15ß	−0.225 (0.579)	−0.280 (0.865)	1.268 (1.370)	<0.001
17OHPo	0.444 (0.669)	−0.903 (0.381)	1.023 (1.028)	<0.001
17OHPo-5α	0.043 (0.928)	−0.396 (0.788)	0.942 (1.100)	0.009
15ß,17OHPo	0.126 (0.656)	−0.579 (0.353)	1.043 (1.776)	<0.001
PT	0.405 (0.546)	−0.865 (0.453)	1.068 (1.206)	<0.001
THA	0.282 (0.912)	−0.732 (0.438)	1.011 (1.031)	<0.001
THF	−0.069 (0.446)	0.043 (1.512)	0.172 (0.756)	0.861
F	−0.177 (0.100)	0.193 (1.629)	0.067 (0.358)	0.562
6ß-OH-F	−0.270 (0.143)	0.360 (1.584)	−0.103 (0.271)	0.180
20α DHF	−0.126 (0.548)	0.146 (1.451)	0.084 (0.823)	0.724
11-O-An	0.261 (0.616)	−0.864 (0.616)	1.349 (0.728)	<0.001
11-OH-An	0.168 (0.597)	−0.675 (0.493)	1.184 (1.518)	<0.001
THE	0.033 (0.733)	−0.513 (0.673)	1.079 (1.481)	0.001
α-Cl	0.201 (0.654)	−0.478 (1.077)	0.262 (1.156)	0.071
6α-OH-THE	−0.437 (0.357)	0.246 (1.242)	0.344 (1.101)	0.053
1ß-OH-THE	−0.209 (0.088)	−0.093 (0.352)	−0.111 (0.191)	0.332
ß-Cl	0.358 (0.878)	−0.644 (0.585)	0.641 (1.298)	0.001
6α-OH-α-Cl	0.107 (0.707)	−0.464 (0.978)	0.884 (1.216)	0.008
6α-OH-ß-Cl	−0.262 (0.312)	0.135 (1.472)	0.357 (0.926)	0.303
1ß-OH-ß-Cl	−0.164 (0.582)	0.008 (1.330)	0.359 (1.134)	0.512
A5-3ß,17α	0.084 (0.744)	−0.730 (0.556)	1.441 (0.835)	<0.001
DHEA	0.431 (0.978)	−0.720 (0.132)	0.561 (1.300)	<0.001
Adiol	−0.044 (0.978)	−0.127 (0.631)	0.365 (1.708)	0.561
16α-OH-DHEA	0.424 (0.800)	−0.847 (0.264)	0.840 (1.255)	<0.001
16ß-OH-DHEA	−0.090 (0.540)	−0.391 (1.210)	1.065 (0.816)	0.003
15ß,16αOH-DHEA	−0.390 (0.276)	0.290 (1.316)	0.122 (1.052)	0.100
16-O-A5D	0.461 (0.717)	−0.901 (0.192)	0.909 (1.216)	<0.001
A5T-16α	0.230 (0.647)	−0.750 (0.282)	1.097 (1.573)	<0.001
A5T-16ß	0.315 (0.619)	−0.798 (0.408)	0.598 (1.368)	<0.001
16α,18OH-DHEA	−0.375 (0.431)	0.183 (1.338)	0.405 (0.962)	0.108
A5-3ß16α17ß18-tetrol	−0.412 (0.512)	0.284 (1.290)	0.343 (1.014)	0.067
A5-3ß15ß16α17ß-tetrol	−0.329 (0.777)	0.245 (1.276)	0.284 (0.705)	0.174
16OHP5o	−0.372 (0.381)	0.028 (1.332)	0.888 (0.861)	0.015
21OHP5o	−0.166 (0.453)	−0.530 (0.816)	1.562 (0.978)	<0.001
P5-3ß20α21-triol	0.235 (1.002)	−0.723 (0.298)	1.014 (0.914)	<0.001
P5-tetrol	−0.039 (0.570)	−0.601 (0.784)	1.446 (1.034)	<0.001
11ß-HSD *	−0.366 (0.490)	0.479 (1.349)	−0.056 (0.821)	0.043

* ratio of 11-OH/11-OXO metabolites.

**Table 7 biomolecules-14-00235-t007:** Distinctive metabolic characteristics of the generated 3 clusters in early preterm infants at the 3rd day and 2nd week of life.

	Cluster 1	Cluster 2	Cluster 3
*n*			
Distinctive metabolic characteristics (day 3)	Mild Elevation of C_21_-steroids (particular 17-OH-progesterone, and cortisone metabolites), and C_19_-steroids (DHEA-metabolites)	No specific abnormality, lowest excretion in general	Highest elevation of C_21_-steroids (17-OH-pregnenolone metabolites, and cortisone metabolites) and C_19_-steroids (DHEA-metabolites)
Distinctive metabolic characteristics (week 2)	Mild Elevation of C_21_-steroids (particular 17-OH-progesterone, cortisol and cortisone metabolites), pregnenolone-metabolites and C_19_-steroids (DHEA-metabolites) compared to cluster 2 but slightly lower than cluster 1 at day 3	No specific abnormality, lowest excretion in general but higher compared to day 3	Elevation of C21-steroids (particular cortisone metabolites) and C_19_-steroids (DHEA-metabolites) compared to cluster 1 and 2 but lower compared cluster 3 at day 3

**Table 8 biomolecules-14-00235-t008:** Clinical phenotype of well and ill early preterm infants according to cluster division (median (min-max)).

	Cluster 1		Cluster 2		Cluster 3		*p*-Value
Early preterm	Well (*n* = 6)	Ill (*n* = 18)	Well (*n* = 3)	Ill (*n* = 11)	Well (*n* = 6)	Ill (*n* = 7)	
GA	29 (25.6–30)	27.6 (24.3–29.9)	28.7 (28.6–28.7)	27.9 (25.1–30)	27.9 (24–29.6)	27.0 (26.3–27.9)	0.50
BW	1155 (700–1460)	1046 (460–1650)	1150 (660–1240)	850 (580–1180)	910 (640–1590)	1050 (880–1340)	0.16
Sex (female)	3/6	11/18		6/11	2/6	2/7	
SGA	0/6	2/18		3/11	1/6	0/7	0.17
Apgar 5 min	9 (8–10)	9 (5–10)	9	8 (7–9)	9.5 (8–10)	8 (7–9)	0.31
C-section		15/18		11/11	2/6	7/7	0.07
pH	7.35 (7.30–7.39)	7.31 (7.15–7.46)	7.26 (7.24–7.4)	7.29 (7.12–7.39)	7.31 (7.14–7.42)	7.35 (7.22–7.37)	0.59
Prenatal Steroids	1/6	10/18		5/11	5/6	4/7	0.32
SNAP D3	8.5 (3–17)	17 (4–33)	6 (4–7)	19 (6–33)	9.5 (2–16)	25.5 (11–31)	0.81
SNAP W2	5 (2–7)	5.5 (2–23)	5 (4–5)	5 (2–28)	5 (3–14)	8 (5–11)	0.69
RDS >II°	0	4/18	0	7/11	0	4/7	0.49
Infection	0	15/18	0	11/11	0	5/7	0.10
IVH > 2	0/6	6/18	0	1/11	0	0	0.08
NEC	0	2/18	0	1/11	0	1/7	0.80
ROP	1/6	1/18		1/11	1/6		0.23
PVL	0	1/18			0	1/7	0.57

**Table 9 biomolecules-14-00235-t009:** Z-transformed daily excretion rates of steroid metabolites of well and ill early preterm infants according to cluster division (Mean (and SD)) and the 3rd day of life.

	Cluster 1	Cluster 2	Cluster 3	*p*
*n*	24 (47.1%)	14 (27.5%)	13 (25.5%)	
THS	0.533 (0.991)	−0.638 (0.322)	−0.296 (1.018)	<0.001
15ß,17OHP5o	−0.059 (0.957)	−0.662 (0.377)	0.821 (1.004)	<0.001
P5-tetrol-15ß	0.061 (0.826)	−0.860 (0.575)	0.814 (0.956)	<0.001
17OHPo	0.610 (1.090)	−0.508 (0.570)	−0.578 (0.360)	<0.001
17OHPo-5α	0.645 (0.883)	−0.417 (0.730)	−0.742 (0.680)	<0.001
15ß,17OHPo	0.454 (1.081)	−0.513 (0.534)	−0.285 (0.901)	0.006
PT	0.645 (0.904)	−0.217 (0.803)	−0.957 (0.156)	<0.001
THA	0.499 (1.189)	−0.537 (0.583)	−0.343 (0.358)	0.002
THF	0.371 (1.172)	−0.541 (0.234)	−0.102 (0.920)	0.020
F	0.262 (1.316)	−0.396 (0.070)	−0.058 (0.719)	0.143
6ß-OH-F	−0.052 (0.907)	−0.324 (0.069)	0.445 (1.497)	0.128
20α DHF	0.276 (1.248)	−0.418 (0.293)	−0.060 (0.853)	0.114
11-O-An	0.490 (0.913)	−0.414 (0.679)	−0.460 (1.076)	0.003
11-OH-An	0.678 (0.879)	−0.485 (0.516)	−0.730 (0.790)	<0.001
THE	0.335 (0.896)	−0.997 (0.685)	0.455 (0.715)	<0.001
α-Cl	0.336 (1.245)	−0.566 (0.526)	−0.012 (0.544)	0.024
6α-OH-THE	−0.283 (0.581)	−0.578 (0.157)	1.145 (1.235)	<0.001
1ß-OH-THE	−0.095 (0.700)	−0.334 (0.180)	0.536 (1.653)	0.061
ß-Cl	0.577 (1.215)	−0.437 (0.234)	−0.595 (0.084)	<0.001
6α-OH-α-Cl	0.546 (1.222)	−0.548 (0.307)	−0.419 (0.255)	<0.001
6α-OH-ß-Cl	0.138 (1.070)	−0.795 (0.134)	0.601 (0.888)	<0.001
1ß-OH-ß-Cl	0.121 (1.014)	−0.887 (0.135)	0.732 (0.805)	<0.001
A5-3ß,17α	0.272 (1.041)	−0.722 (0.384)	0.277 (1.044)	0.005
DHEA	0.254 (0.886)	−0.539 (0.347)	0.112 (1.432)	0.052
Adiol	−0.209 (0.890)	−0.233 (0.618)	0.637 (1.282)	0.026
16α-OH-DHEA	0.341 (1.033)	−0.534 (0.441)	−0.055 (1.158)	0.029
16ß-OH-DHEA	−0.064 (0.568)	−0.564 (0.144)	0.725 (1.606)	0.002
15ß,16α -OH-DHEA	−0.431 (0.419)	−0.540 (0.375)	1.378 (0.953)	<0.001
16-O-A5D	0.515 (1.035)	−0.554 (0.491)	−0.354 (0.912)	0.001
A5T-16α	0.321 (0.839)	−0.380 (0.881)	−0.184 (1.250)	0.082
A5T-16ß	0.163 (1.003)	−0.371 (0.334)	0.098 (1.376)	0.265
16α,18OH-DHEA	−0.216 (0.749)	−0.573 (0.202)	1.015 (1.206)	<0.001
A5-3ß16α17ß18-tetrol	−0.330 (0.499)	−0.527 (0.285)	1.176 (1.250)	<0.001
A5-3ß15ß16α17ß-tetrol	−0.245 (0.722)	−0.608 (0.372)	1.107 (1.066)	<0.001
16OHP5o	−0.217 (0.604)	−0.682 (0.263)	1.136 (1.165)	<0.001
21OHP5o	−0.023 (0.661)	−0.911 (0.450)	1.024 (1.003)	<0.001
P5-3ß20α21-triol	0.379 (0.922)	−0.620 (0.335)	−0.031 (1.296)	0.009
P5-tetrol	0.014 (0.723)	−0.711 (0.206)	0.740 (1.392)	<0.001
11ß-HSD *	0.178 (1.344)	−0.292 (0.303)	−0.013 (0.666)	0.384

* ratio of 11-OH/11-OXO metabolites.

## Data Availability

The datasets acquired during and/or analyzed during the current study are available from the corresponding author upon reasonable request.

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
