# Peer review of "Steroid Metabolomic Signature in Term and Preterm Infants"

_biomolecules, 2024, doi:10.3390/biom14020235_

Round 1

Reviewer 1 Report

Comments and Suggestions for Authors

The authors of the manuscript “Steroid metabolomic signature in term and preterm infants” present the results of the study on the identification of steroidal fingerprints within the steroid metabolome of in-term and preterm infants. In general, the manuscript is well-written, easy to follow, and with clear conclusions. The limitation is the list of chemicals. For clarification, the authors should consider the identification of tested compounds by CAS number in SI. The study itself is interesting and as stated by the authors has limitations with the number of infants tested and the list of compounds identified. Yet more of these studies are needed, I do agree with the authors.

I do have only minor comments:

Abstract:
- please do consider the revision of the abstract. It is complicated and not easy to follow. You have been using many “new terms” without explanation.
For example,
- line 19 – C19 steroids from the fetal zone – it is not explained what is the “fetal zone”.
- C19 steroids are referred to in line 19 as DHEA-metabolites while in line 50 as androgens. For a steroid chemist, it is clear, for a nonsteroidal expert, it is not.
- C21 steroids are not defined
- term infants are not defined
- line 24 – The excretion rates … is this sentence related to all 3 clusters or not?

Main text:
- lines 162 and 163 … only abbreviations are used … no explanation. Is this your calculation or is it related to some previous literature?
- line 179 – abbreviations for (20,21)
- line 247 – Capitalize Table and Figure – revision through the whole text is needed
- line 247 – table 2 and table 3 – maybe, it should be 2C and 3C?
- Figure 3 – it is a low resolution and the metabolites are not readable
- line 278 – the sentence does not make sense. The word “more” should be deleted?
- line 346 – dehydroepiandrosterone SPACE sulfate

A major revision of the nomenclature of steroids in SI Table 1 is needed. I have been doing steroid chemistry for almost 20 years and to be honest, I was lost. First, double abbreviations are used. It makes sense for DHEA, for example. But not for all compounds. For example, THF is 5β-Pregnane-3α,11β,17α,21-tetrol-20-on (TH-Cortisol). I do understand that TH means tetrahydro, but why is there a double name TH-Cortisol?
Second, in case the skeleton is pregnane, it is not needed to name 17 substituents with alpha as the endogenous compounds can not be 17beta.
Next, in case C-20 has a hydroxyl group, you can not use alpha or beta – it is not correct, use proper R/S system of naming.
Next, abbreviations do not make sense in general usage – for example, 15ß-OH-Po  - OH most likely means hydroxyl at C-15. However, this is not used systematically for all compounds.
Next, what is abbreviation “O” ?
Next, “Cl” is in chemistry chlorine … you have been using is cortolon.
Next, “THF” is in chemistry abbreviation for tetrahydrofuran.
Next,  11-O-An 5α-anedrostane-3α-ol-11,17-dione – should be androstane
Next, there is no system in naming. For example 15ß,17αOH-P5olon versus Po-5α,3α – numbers first, then name.
Next, “on” means keto group. It should -ONE

Draw the structures in Scifinder, identify CAS numbers and trivial names, and follow the nomenclature generally used for skeletons. Let me illustrate the options for modification following IUPAC nomenclature and priorities in nomenclature of substituents. The substituent with the highest priority should be at the end. The substituent(s) with lower priority should be at the beginning (HYDROXY-skeleton-KETONE vs. skeleton-HYDROXY)

Example 1:
Abbreviation: P5-tetrol-15ß
Urinary steroid metabolites: 5-Pregnene-3β,15β,17α,20α-tetrol
Modified:
Abbreviation: 3β,15β,17,20(S)-P5-tetrol
Urinary steroid metabolites: Pregn-5-ene-3β,15β,17,20(S)-tetrol

Example 2:
Abbreviation: 20α-DHF
Trivial name: 20α-Dihydrocortisol
Urinary steroid metabolite: 11β,17,20(S),21-Tetrahydroxy-Pregn-4-en-3-one

Author Response

The authors of the manuscript “Steroid metabolomic signature in term and preterm infants” present the results of the study on the identification of steroidal fingerprints within the steroid metabolome of in-term and preterm infants. In general, the manuscript is well-written, easy to follow, and with clear conclusions. The limitation is the list of chemicals. For clarification, the authors should consider the identification of tested compounds by CAS number in SI. The study itself is interesting and as stated by the authors has limitations with the number of infants tested and the list of compounds identified. Yet more of these studies are needed, I do agree with the authors.

Response: We thank the reviewer for the positive reception of our revised manuscript.

I do have only minor comments:

Abstract:
- please do consider the revision of the abstract. It is complicated and not easy to follow. You have been using many “new terms” without explanation. 
For example, 
- line 19 – C19 steroids from the fetal zone – it is not explained what is the “fetal zone”.

- C19 steroids are referred to in line 19 as DHEA-metabolites while in line 50 as androgens. For a steroid chemist, it is clear, for a nonsteroidal expert, it is not. 
- C21 steroids are not defined
- term infants are not defined

- line 24 – The excretion rates … is this sentence related to all 3 clusters or not?

Response: We thank the reviewer for these comments. Explanations and definitions were included and, also with respect to 2ndReviewer, the abstract reads now: `In addition to glucocorticoids, it has been hypothesized that C19-steroids (DHEA-metabolites) from the fetal zone of the adrenal gland may play a role as endogenous neuroprotective steroids. In 39 term-born (<37 weeks gestational age), 42 preterm (30-36 weeks) and 51 early preterm (< 30 weeks) infants 38 steroid metabolites were quantified by GC-MS in 24-hour urinary samples.´ … Overall, the excretion rates of C21-steroids (glucocorticoid precursors, cortisol, and cortisone metabolites) were low (< 99 mg/kg body weight/d) whereas the excretion rates of C19-steroids were up to 10times higher.

Furthermore, additional explanation was added to the Introduction section to keep the Abstract within the word count: `The human adult adrenal cortex produces mineralocorticoids, glucocorticoids (C21-steroids which are characterised by 2 methyl groups and a side chain with 2 carbon atoms on the sterane skeleton), and adrenal androgens (C19-steroids which are formed by splitting off the side chain of the C-21 steroids with the 2 carbon atoms). In the fetus, an additional inner zone, the fetal zone, produces extremely high amounts of C19-steroids, mainly dehydroepiandrosterone (DHEA) metabolites.´

Main text:
- lines 162 and 163 … only abbreviations are used … no explanation. Is this your calculation or is it related to some previous literature? 

Response: The SI-unit for pressure was used: (kilo)pascal. The press was self-constructed by the precision mechanics workshop for medical devices of the University of Giessen on the basis of a car jack with known Pressure force. The sentence reads now: ` Urine was extracted by compressing the diaper using a hydraulic press applying a maximum of 120 kilopascal(kPa)/cm2

- line 179 – abbreviations for (20,21)

Response: We apologize for the wrong reference. It was corrected accordingly (20,21).

- line 247 – Capitalize Table and Figure – revision through the whole text is needed

Response: Done.

- line 247 – table 2 and table 3 – maybe, it should be 2C and 3C?

Response: Because the tables present both clinical and metabolic data, the sentence was not changed: `Their descriptive data are presented in Table 2 (preterm) and Table 3 (early preterm).´

- Figure 3 – it is a low resolution and the metabolites are not readable

Response: We divided and reorganized Figure 3 in Fig. 3A (preterm) and 3B (early preterm) to enhance resolution.

- line 278 – the sentence does not make sense. The word “more” should be deleted?

Response: We agree with the reviewer and the sentence reads now: `In the second week, considerably overlap in the metabolic clusters was observed while the median (min-max) SNAP score decreased from 5 (0-15) at the 3rd day to 2 (0-7) in the 2nd week.´ Furthermore, in this context a following sentence was revised, too: `In the second week, the metabolic clusters showed overlap but to a lesser extent compared to the more mature preterm infants as the median (min-max) SNAP score decreased from 15 (2-33) at the 3rd day to 5 (2-28) in the 2nd week.´

- line 346 – dehydroepiandrosterone SPACE sulfate

Response: Done.

A major revision of the nomenclature of steroids in SI Table 1 is needed. I have been doing steroid chemistry for almost 20 years and to be honest, I was lost. First, double abbreviations are used. It makes sense for DHEA, for example. But not for all compounds. For example, THF is 5β-Pregnane-3α,11β,17α,21-tetrol-20-on (TH-Cortisol). I do understand that TH means tetrahydro, but why is there a double name TH-Cortisol? 

Response: We agree that steroid nomenclature can be complex and confusing. We further agree with the reviewer that the IUPAC rules are actually authoritative. However, the IUPAC nomenclature is too complex for clinical endocrinology laboratory work, so a system of abbreviations and trivial names has been retained for reasons of practicality. In our work we therefore follow the practice and tradition of the school of Shackleton (e.g. J Steroid Biochem Mol Biol 1993;45:127), who is regarded as the forefather of clinical mass spectrometric steroid analysis and was the mentor of the senior author (SAW). Its abbreviations and trivial names have proven beneficial for us and others over the last 20-30 years in our daily lab and clinical work. Nevertheless we have tried to improve and further clarify steroid nomenclature in the manuscript (see revised Table S1).

Second, in case the skeleton is pregnane, it is not needed to name 17 substituents with alpha as the endogenous compounds can not be 17beta. 

Response: Reply: we agree and have made changes accordingly (see revised Table S1).

Next, in case C-20 has a hydroxyl group, you can not use alpha or beta – it is not correct, use proper R/S system of naming. 

Response: The reviewer is formally absolutely right. However, in this matter we prefer to stick to the practice of Shackleton and also to the naming convention of the Steraloids Company (see below).

Next, abbreviations do not make sense in general usage – for example, 15ß-OH-Po  - OH most likely means hydroxyl at C-15. However, this is not used systematically for all compounds.

Response: Yes, 15ß-OH-Po means hydroxyl at C-15. The mention of the OHgroup is to discern the many pregnanes. Likewise we adapted the names of other pregnanes.

Next, what is abbreviation “O” ?

Response: ..stands for ketone.

Next, “Cl” is in chemistry chlorine … you have been using is cortolon. 

Response: …in the context of urinary steroids Cl stands for cortolone (see Shackleton).

Next, “THF” is in chemistry abbreviation for tetrahydrofuran. 

Response: …likewise, in the context of urinary steroids THF stands for tetrahydrocortisol (see Shackleton).

Next,  11-O-An 5α-anedrostane-3α-ol-11,17-dione – should be androstane

Response: Thank you, has been corrected.

Next, there is no system in naming. For example 15ß,17αOH-P5olon versus Po-5α,3α – numbers first, then name. 

Response: Thank you, has been corrected

Next, “on” means keto group. It should -ONE

Response: Thank you, has been corrected.

Draw the structures in Scifinder, identify CAS numbers and trivial names, and follow the nomenclature generally used for skeletons. Let me illustrate the options for modification following IUPAC nomenclature and priorities in nomenclature of substituents. The substituent with the highest priority should be at the end. The substituent(s) with lower priority should be at the beginning (HYDROXY-skeleton-KETONE vs. skeleton-HYDROXY)

Response: Thank you for your referral to the IUPAC nomenclature for systematic names. We are aware of the IUPAC rules. As already stated above, IUPAC nomenclature is complex and thus not useful in daily steroid routine practice. The system we follow is that of Shackleton and the Steraloids Company (see naming convention by Steraloids, www.steraloids.com). Steraloids is the leading company for providing pure standard material. Their naming convention is very logical and allows quick orientation about the structure of the molecule. We have been successfully following this convention in all our publications for the last 30 years!

Reviewer 2 Report

Comments and Suggestions for Authors

This study offers valuable insights into the intricate relationship between steroid metabolism and neonatal outcomes among preterm infants, shedding light on potential avenues for further research. It raises intriguing questions about the neuroprotective roles of C19-steroids and their implications in neonatal care. The study underscores the importance of considering individual steroidal profiles to comprehend and address the health challenges confronting premature infants.

However, there are several noteworthy points that warrant examination:

1. Introduction:

The introduction contains speculative statements regarding the neuroprotective role of fetal zone steroids (FZSs). To enhance clarity, it would be advantageous to present these statements explicitly as hypotheses rather than as definitive conclusions.

2. Patients and Methods:

a) Not recording the total number of patients screened and parental refusals represents a significant oversight, as it impacts our understanding of the study's representativeness and the potential for selection bias.

b) The method of calculating urine volume based on diaper weight may introduce errors due to factors such as evaporation or inaccuracies in initial diaper weight.

c) The decision to analyze steroid metabolites exclusively on the 3rd day and 2nd week of life might overlook critical variations at other times, potentially limiting the study's scope.

d) While employing Bonferroni adjustment for multiple testing is considered good practice, it is known for its conservatism, which could lead to Type II errors (failure to detect a difference when one exists).

3. Results:

a) Despite observing varying levels of steroid metabolites across clusters, the study does not establish significant correlations between these levels and clinical outcomes. For example, it does not report any significant differences in disease severity or type among clusters. Conducting additional statistical analyses to explore potential correlations between steroid metabolite levels and specific clinical outcomes would be beneficial. Consideration of more sophisticated statistical models capable of handling complex relationships and interactions between variables, along with a combination of statistical techniques such as Gaussian Mixture Models (GMMs), Hierarchical Clustering, PCA, and Longitudinal Data Analysis, could provide a more comprehensive and robust analysis in this study.

b) The substantial overlap in metabolic clusters, particularly in the second week, suggests a lack of clear differentiation in the steroid metabolomic profiles among the different groups of infants. This implies that the clusters may not be as distinct as initially hypothesized.

c) The ROC analysis failed to identify any of the selected metabolites as reliable biomarkers for distinguishing between ill and well early preterm infants. This is a significant outcome, indicating that the selected metabolites may not be as clinically informative as anticipated.

4. Discussion:

a) The discussion alludes to potential clinical implications of the findings, such as using steroid metabolome analysis to guide therapy in early preterm infants. However, it's important to underscore that the study does not provide conclusive evidence of the practical utility of this approach in a clinical setting. The discussion should exercise caution in overstating the clinical significance of the results without additional concrete evidence from larger-scale studies or clinical trials.

b) While the study explores the relationship between steroid metabolites and disease severity, it acknowledges the complexity of the interplay between various factors. It recognizes that gestational age and disease severity alone do not account for all the variations observed in the metabolome clusters. This underscores the pressing need for further research to gain a comprehensive understanding of the multifaceted nature of neonatal health.

Author Response

This study offers valuable insights into the intricate relationship between steroid metabolism and neonatal outcomes among preterm infants, shedding light on potential avenues for further research. It raises intriguing questions about the neuroprotective roles of C19-steroids and their implications in neonatal care. The study underscores the importance of considering individual steroidal profiles to comprehend and address the health challenges confronting premature infants.

Response: We thank the reviewer for the positive reception of our revised manuscript.

However, there are several noteworthy points that warrant examination:

  1. Introduction:

The introduction contains speculative statements regarding the neuroprotective role of fetal zone steroids (FZSs). To enhance clarity, it would be advantageous to present these statements explicitly as hypotheses rather than as definitive conclusions.

Response: Following the reviewer´s suggestion, we changed the last sentence of the paragraph on the hypothetical neuroprotective role of fetal zone steroids: `We therefore hypothesised from the experimental data that the persistently high concentrations of FZSs might lead to a transient endogenous neuroprotection in human preterm infants [14].´

  1. Patients and Methods:
  2. a) Not recording the total number of patients screened and parental refusals represents a significant oversight, as it impacts our understanding of the study's representativeness and the potential for selection bias.

Response: We thank the reviewer for this comment and added this paragraph to the Patients and Methods section: `We did not expect a major selection bias because the overall acceptance of the study was good because there was no intervention or invasive procedure. Only the diapers were collected. Therefore, the total number of patients screened and parental refusals were not recorded with the exception of the early preterm group because this gestational age group represented only 10 percent of neonatal patients treated. The 67 preterm infants with a gestational age of less than 30 weeks were admitted to our neonatal unit between July 2001 and September 2002. Sixty-one of them matched the study criteria. Reasons for exclusion were postnatal steroid therapy (n=5) and chromosomal aberration (trisomy 21, n=1).´

  1. b) The method of calculating urine volume based on diaper weight may introduce errors due to factors such as evaporation or inaccuracies in initial diaper weight.

Response: We thank the reviewer for this comment. Using diaper for calculating urine volume and collect urinary metabolites was already investigated in detail (1. Ahmad T, Vickers D, Campbell S, Coulthard MG, Pedler S 1991 Urine collection from disposable nappies. Lancet 338:674–676 and Roberts SB, Lucas A 1985 Measurement of urinary constituents and output using disposable napkins. Arch Dis Child 60:1021–1024). It was shown that collecting urine samples from compressible nappies, taking care to minimize evaporation (Gouyon JB, Sonveau N, d’Athis P, Chaillot B 1994 Accuracy of urine output measurement with regular disposable nappies. Pediatr Nephrol 8:88–90) and correcting for losses by weighing is a suitable alternative for measuring accurately 24-h urine output and determining several constituents of urine. Furthermore, this method does not interfere with the nursing care and is completely noninvasive and nonstressful. In our study, the diapers were closed properly to limit evaporation, even when the infant underwent phototherapy. This issue was described and discussed in detail in our publication on the method to collect urine for measuring for urinary steroids in preterm infants (Heckmann, M.; Hartmann, M.F.; Kampschulte, B.; Gack, H.; Bodeker, R.H.; Gortner, L.; Wudy, S.A. Assessing cortisol production in preterm infants: do not dispose of the nappies. Pediatr Res 2005, 57, 412-418.).

  1. c) The decision to analyze steroid metabolites exclusively on the 3rd day and 2nd week of life might overlook critical variations at other times, potentially limiting the study's scope.

Response: We agree with the reviewer. Samples were also collected at other times (first, second, third, and fifth day of life, and at weekly intervals during the first month of life and then monthly up to discharge). However, the decision to concentrate on day 3 and the 2nd week was based on these reasons as stated in the manuscript: `Timing was chosen to maximize the number of urine samples available during the first week of life when disease severity was high and a later time point after recovery from acute illness (20,21). Hence, the time points 3rd day of life and 2nd week of life were selected. However, this analysis is limited to preterm infants, as the term infants were predominantly discharged before the 2nd week of life.´

We added the reviewers remark to the limitations of the study: `Furthermore, analysing steroid metabolites exclusively on the 3rd day and 2nd week of life might overlook critical variations at other times.´

  1. d) While employing Bonferroni adjustment for multiple testing is considered good practice, it is known for its conservatism, which could lead to Type II errors (failure to detect a difference when one exists).

Response: In our analyses we used the Bonferroni adjustment for multiple testing. Since this correction is quite conservative, we might have overlooked some weaker associations of the metabolites.

  1. Results:
  2. a) Despite observing varying levels of steroid metabolites across clusters, the study does not establish significant correlations between these levels and clinical outcomes. For example, it does not report any significant differences in disease severity or type among clusters. Conducting additional statistical analyses to explore potential correlations between steroid metabolite levels and specific clinical outcomes would be beneficial. Consideration of more sophisticated statistical models capable of handling complex relationships and interactions between variables, along with a combination of statistical techniques such as Gaussian Mixture Models (GMMs), Hierarchical Clustering, PCA, and Longitudinal Data Analysis, could provide a more comprehensive and robust analysis in this study.

Response: As recommended by the reviewer, we used a principle component analysis (PCA) to define the three clusters in each of the groups (early preterms, preterms and terms). The detailed procedure is described in the Statistical Methods section. Furthermore, following the reviewers request, we have performed additional analyses to the Result section: `The odds ratio for severe IVH was 4.33 (CI 0.46-40.5; p= 0.198) for cluster 1 versus cluster 2.´ Furthermore, we extend the ROC analysis (see c)) for the clinical outcome IVH: `Furthermore, ROC analysis did not identified any of the selected metabolites to dis-criminate between severe IVH (> 2) and less severe or no IVH in early preterm infants (Number of observations = 51): ß-Cl (area under ROC curve = 0.6104), 6α-OH-α-Cl (0.5357), 6α-OH-ß-Cl (0.5065), 16α-OH-DHEA (0.5812), and A5-3ß16α17ß18-tetrol (0.4675). ´

  1. b) The substantial overlap in metabolic clusters, particularly in the second week, suggests a lack of clear differentiation in the steroid metabolomic profiles among the different groups of infants. This implies that the clusters may not be as distinct as initially hypothesized.

Response: We agree with the reviewer and added this to the corresponding paragraph of the Discussion section: `In the second week of life, severity of illness measured by SNAP decreased in the majority of preterm and early preterm infants. At the same time, the metabolic clusters converge. The substantial overlap in metabolic clusters in the second week suggests a lack of clear differentiation in the steroid metabolomic profiles among the different groups of infants. Although there were significant differences in steroid metabolism between clusters in the immediate postnatal phase at day 3, these are only partly associated with the clinical phenotype, i.e. disease severity in term infants and incidence of severe cerebral hemorrhage in the early preterm group.´

  1. c) The ROC analysis failed to identify any of the selected metabolites as reliable biomarkers for distinguishing between ill and well early preterm infants. This is a significant outcome, indicating that the selected metabolites may not be as clinically informative as anticipated.

Response: As recommended by the reviewer, we have performed additional analyses (see a)). However, these additional analyses also failed to detect a reliable biomarker to distinguish between early preterm infants with and without severe intracranial hemorrhage.

With respect to the reviewer´s comment, the Discussion section and the Conclusion has been amended accordingly: `Biomarker analysis failed to identify a metabolite which discriminate between well and ill early preterm infants and those with and without severe intracranial hemorrhage.´ … `Furthermore, analysis of selected steroid metabolites failed to identify reliable biomarkers for distinguishing between ill and well early preterm infants.´

  1. Discussion:
  2. a) The discussion alludes to potential clinical implications of the findings, such as using steroid metabolome analysis to guide therapy in early preterm infants. However, it's important to underscore that the study does not provide conclusive evidence of the practical utility of this approach in a clinical setting. The discussion should exercise caution in overstating the clinical significance of the results without additional concrete evidence from larger-scale studies or clinical trials.

Response: We agree with the reviewer and added this sentence to the Discussion section: `The study does not provide conclusive evidence of the practical utility of this approach in a clinical setting.´

  1. b) While the study explores the relationship between steroid metabolites and disease severity, it acknowledges the complexity of the interplay between various factors. It recognizes that gestational age and disease severity alone do not account for all the variations observed in the metabolome clusters. This underscores the pressing need for further research to gain a comprehensive understanding of the multifaceted nature of neonatal health.

Response: Following the clear statement of the reviewer, the Conclusion was rewritten: `In conclusion, while the study explores the relationship between steroid metabolites and disease severity, it acknowledges the complexity of the interplay between various factors. It recognizes that gestational age and disease severity alone do not account for all the variations observed in the metabolome clusters.´